# Creating a Millennial-Long Chronology in Northern Canada: Dendroarchaeological Dating of the Moose Horn Pass Caribou Fence (KjRx-1), Mackenzie Mountains, NT

**Gary Beckhusen** [1,2], **Glenn Stuart** [3], **Leon Andrew** [4], **Glen MacKay** [5], **Thomas Andrews** [5] and **Colin Laroque** [1,*]

1   Mistik Askiwin Dendrochronology Laboratory, University of Saskatchewan, 51 Campus Drive, Saskatoon, SK S7N 5A8, Canada; gary.beckhusen@cannorth.com
2   Canada North Environmental Services, 211 Wheeler Street, Saskatoon, SK S7P 0A4, Canada
3   Department of Archaeology and Anthropology, University of Saskatchewan, 55 Campus Drive, Saskatoon, SK S7N 5B1, Canada; glenn.stuart@usask.ca
4   Tulita Dene Band P.O. Box 475, Norman Wells, NT X0E 0V0, Canada; lamountaindene@theedge.ca
5   Prince of Wales Northern Heritage Centre, 4750 48th Street, Yellowknife, NT X1A 2L9, Canada; glen_mackay@gov.nt.ca (G.M.); tomandr@gmail.com (T.A.)
*   Correspondence: Colin.laroque@usask.ca

**Abstract:** The Moose Horn Pass Caribou Fence site (KjRx-1) consists of three wooden fences located in a remote area of the Mackenzie Mountains in Canada's Northwest Territories. Situated in the traditional homeland of the Shúhtagot'ine (Mountain Dene), they were used to assist past hunters to harvest northern mountain caribou by channeling multiple animals toward kill zones. The main fence is nearly 800 m in length and terminates in a corral structure after descending from high ground into a valley. The two smaller fences are located north and south of the main fence, and they do not descend into the valley. Standard dendrochronological methods were employed to determine the ages of wood taken from the fence structures. Seventy-five living white spruce (*Picea glauca*) trees in the area were cored to determine the overall tree-ring growth patterns in the local environment. The chronology of living trees was supplemented by the inclusion of 29 standing-dead trees to establish a longer chronology of dated ring widths. Sixty-two of 89 cross-sections cut from the fence timbers were crossdated and added to the overall chronology, which created a well-replicated chronology of ring-widths from 972 to 2016 C.E. The terminal dates of material from the three fence systems suggest that the complex was built from trees that died over a wide temporal period, spanning the years 1314 to 1876 C.E, with clusters of dates between ca. 1420–1480 and 1580–1750 C.E. The millennial-long chronology developed in this study can now be used as a base to assist in dendroarchaeological dating of many more artifacts from the region.

**Keywords:** dendroarchaeology; Mountain Caribou; Mackenzie Mountains; dendrochronology; Mountain Dene; white spruce; Northwest Territories; millennial-long

## 1. Introduction

The Moosehorn Caribou Fence was constructed by ancestral Shúhtagot'ine hunters as part of a complex hunting technology used to harvest northern mountain caribou [1] (Figure 1). The wooden structures were recorded by Government of Northwest Territories (GNWT) archaeologists in 2009 [1]. The fence system was used to guide caribou to a predetermined kill zone for harvest. This efficient method of hunting required an intimate knowledge of the prey and the land and was used by hunter–gatherer societies throughout North America [2–12].

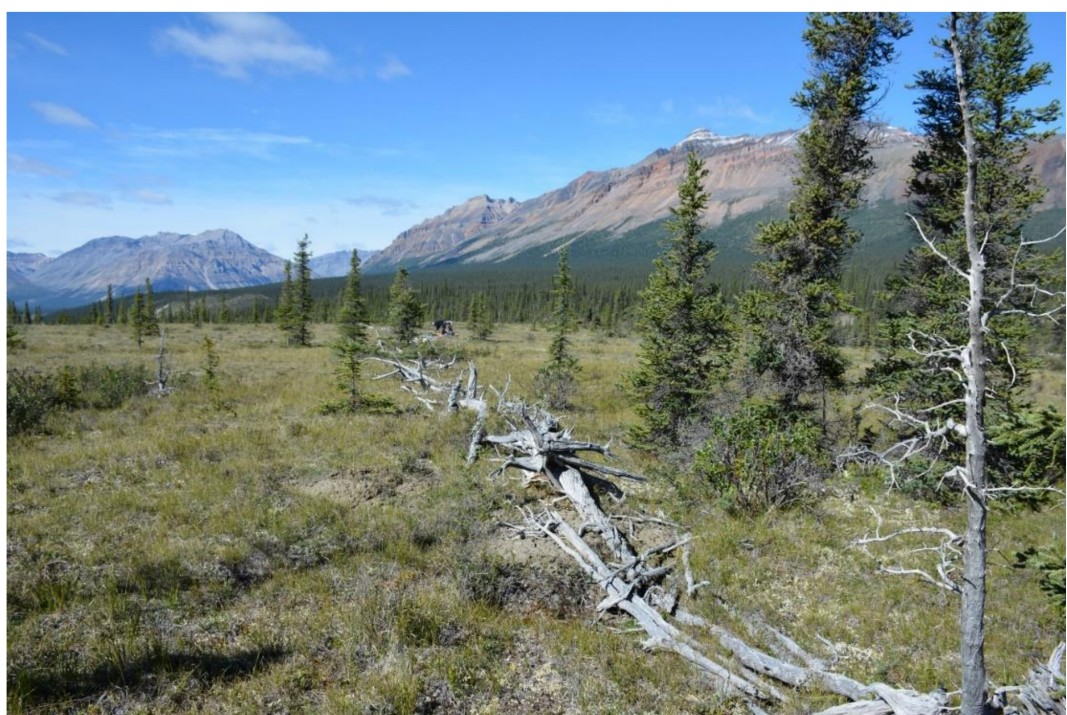

**Figure 1.** A section of the remains of the ≈800 m long mountain caribou fence structure with Stelfox Mountain in the background.

The three fence features at the Moosehorn Caribou Fence site are referred to as main, north, and south fences. The main fence, the largest of the three features, is oriented west to east along the high plateau (Figure 2). On the east side of the plateau, it turns and steeply descends into an adjacent valley, where it terminates in a hook-shaped corral structure. The fence and the corral structure are identified as a single continuous feature. North of the main fence is a smaller feature that runs west to east, following a slightly elevated contour across the plateau. Another fence lies to the south; it is the shortest of the three and is oriented more northwest/southeast than its counterparts. The south fence descends minimally into the valley but it is difficult to determine whether the remains today are an accurate representation of its original extent, or if some portion of the fence has fallen downslope into the valley.

The fence lines were built by using the root masses of trees placed in the fence to prop up adjacent fence timbers (Figure 3). Additional timbers and brush were likely added to this basic structure to create a more substantial barrier. While most of the fence has collapsed, it is estimated to have been approximately 1 m in height. The builders may have made abundant use of unmodified deadfall to build the fences, particularly the timbers that retain their root structures; however, some of the fence timbers have cut marks from steel axes, indicating that they also felled living trees and/or standing dead trees for use in the fence. It is likely that the wood used to build the fences was sourced from the locally available white spruce (*Picea glauca*) trees, given that they are the predominant species in the study area.

Cultural resource managers in the NWT consider the Moose Horn Pass Caribou Fence to be "at risk" of being lost from forest fires. In partnership with the local Shúht-agot'ine people of the Tulit'a Dene Band, a plan was initiated to create a detailed record of the fences and their landscape context in the event it is lost, should a forest fire sweep through the area. Obtaining absolute dates for the fence complex was one of the major goals of the recording process, as this would ultimately allow for a better understanding of the use-history of the fence complex. Dendrochronology was identified as a useful tool to fulfill this objective due to the well-preserved timbers at the site.

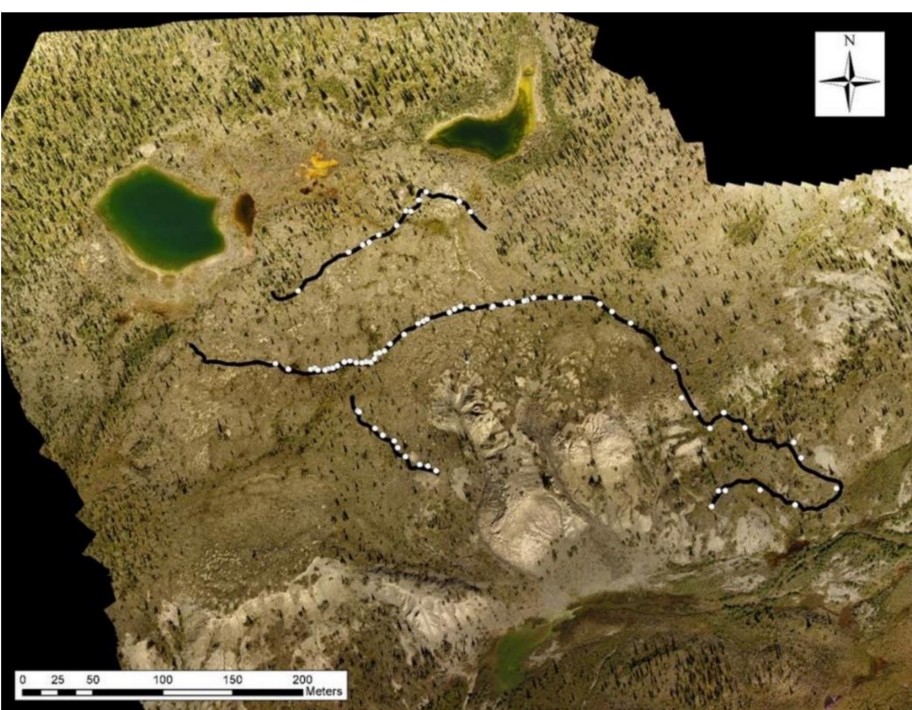

**Figure 2.** A photomosaic of the study site with the three fences indicated by black lines, with locations of where 89 archaeological individual samples were collected marked by white dots along the fences. The longest main fence structure continues over the steep embankment and ends in a hooked corral-like structure in the southeast section of the study site.

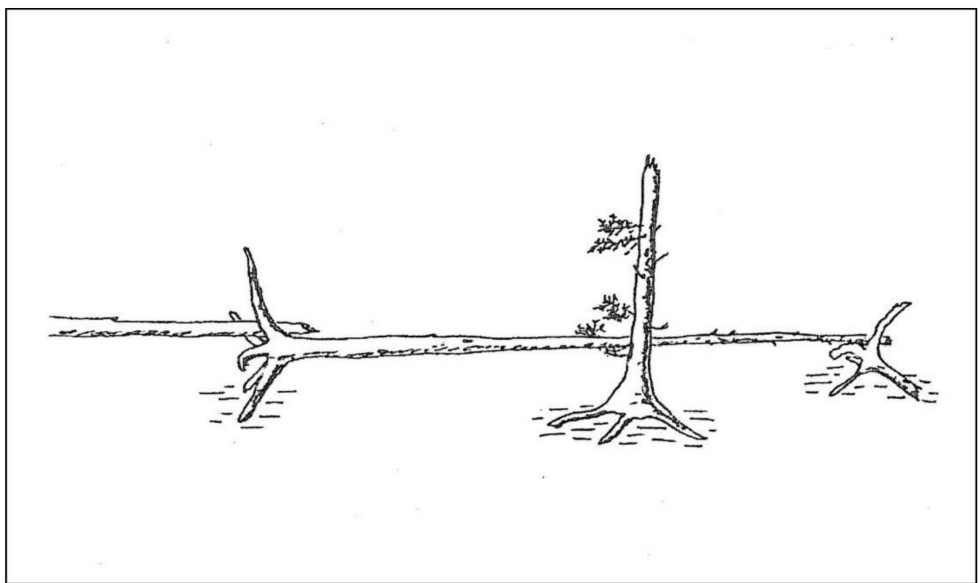

**Figure 3.** A schematic of how the fence was built using wood available to the builders from the surrounding area (after [12]).

Dendroarchaeology is a subfield of dendrochronology in which dendrochronological tree-ring dating techniques are used to date human artifacts and constructions [13–18]. It is accomplished by crossdating annual ring patterns found in archaeological wood to those of living trees in the region where the wood was obtained. Once accomplished, a precise calendar year can be assigned to the last ring formed before a tree was incorporated into an archaeological feature. This level of accuracy can uncover the holistic history of the complex structures that make up the Moose Horn Pass Caribou Fence system.

By applying dendroarchaeological methods, this study seeks to determine the spatial and temporal patterns of use for the three fences. To do so, it is necessary to first construct a master chronology of white spruce growth from living trees in the Moose Horn Pass region. The longer this chronology is, the greater the number of regional archaeological samples that will, theoretically, be able to be placed in time. Consequently, by dating samples from the fence system, information on the timing of initial construction and determining whether use was continuous or episodic, and when the site was last used or repaired, might be established. The spatial and temporal pattern of fence line tree felling dates may also indicate areas of concentrated repair and maintenance, or if the fence system had been redesigned or augmented to form its current layout.

The null hypothesis that was set forth is that the fences were built all at one time: approximately during the time of the Hudson Bay Company meat trade in the region in the early 1900s. Regardless of if this hypothesis is accepted or rejected, the information gathered to answer the question will provide the project's stakeholders, the Shúhtagot'ine People, a more complete picture of the how and when their ancestors utilized the fences in the past.

## 2. Methods

### 2.1. Study Site

The Moose Horn Pass Caribou Fence site is in a remote setting at the base of Stelfox Mountain in the Mackenzie Mountains of Canada's Northwest Territories (Figure 4). The general area around the site is known locally as Moosehorn Pass. The topography is rugged with a change in elevation from valley bottom to the summit of Stelfox Mountain being 1270 m (1070–2340 masl). The fence is situated on a plateau, south of two small water bodies, and north of a steep slope descending approximately 48 m down into an adjacent valley. The valley bottom runs parallel to the plateau and is home to Stelfox Creek between the Moose Horn and Natla Rivers.

Bedrock outcroppings, alluvial fans, and glacial till are found throughout the adjacent alpine environment [21]. Diamictite, overlain by dolostone, quartz sandstone, and shale are the primary components of the Stelfox Mountain geology [22,23]. The site is not easily accessed, and no weather data have been recorded from within the valley. The nearest estimates place its mean annual daily temperature at −6 °C, and there is a February daily high of −30 °C [21]. The area receives approximately 700 mm of moisture in annual precipitation, most of which falls as snow during the winter months [21]. Some amount of snowfall can occur in any month.

The summer growing season is short and lasts approximately 12 to 20 weeks, with plant communities on south-facing aspects faring much better than their north-facing counterparts. The area's tree community is dominated by white spruce, with a few deciduous trees beginning to colonize in lower and more protected elevations. Wild grasses and forbs are present throughout the site area. Stunted shrubs and bushes are present in the vicinity of the water bodies.

### 2.2. Live Tree Sampling

Seventy-five living white spruce trees in the area were sampled at 1.3 m above the ground, and two cores were taken from each tree (n = 150 cores). Most increment cores were obtained from trees in the forested area directly to the north of the fences (Figure 2) using standard 5.1 mm diameter increment borers. Additional samples were extracted from the plateau in the vicinity of the fences and from the slope leading down to the corral of the main fence. Two cores were taken to help eliminate expected aberrations in the ring patterns arising from the effects of the harsh growing conditions. In flat areas, cores were taken at 90° from each other, and on slopes, they were taken at 180° from one another, perpendicular to the slope angle. Cores were transferred to plastic drinking straws, sealed, labeled sequentially, and returned to the laboratory for processing. The calendar year in which living trees are cored provides the necessary anchor for extending the chronology backwards in time.

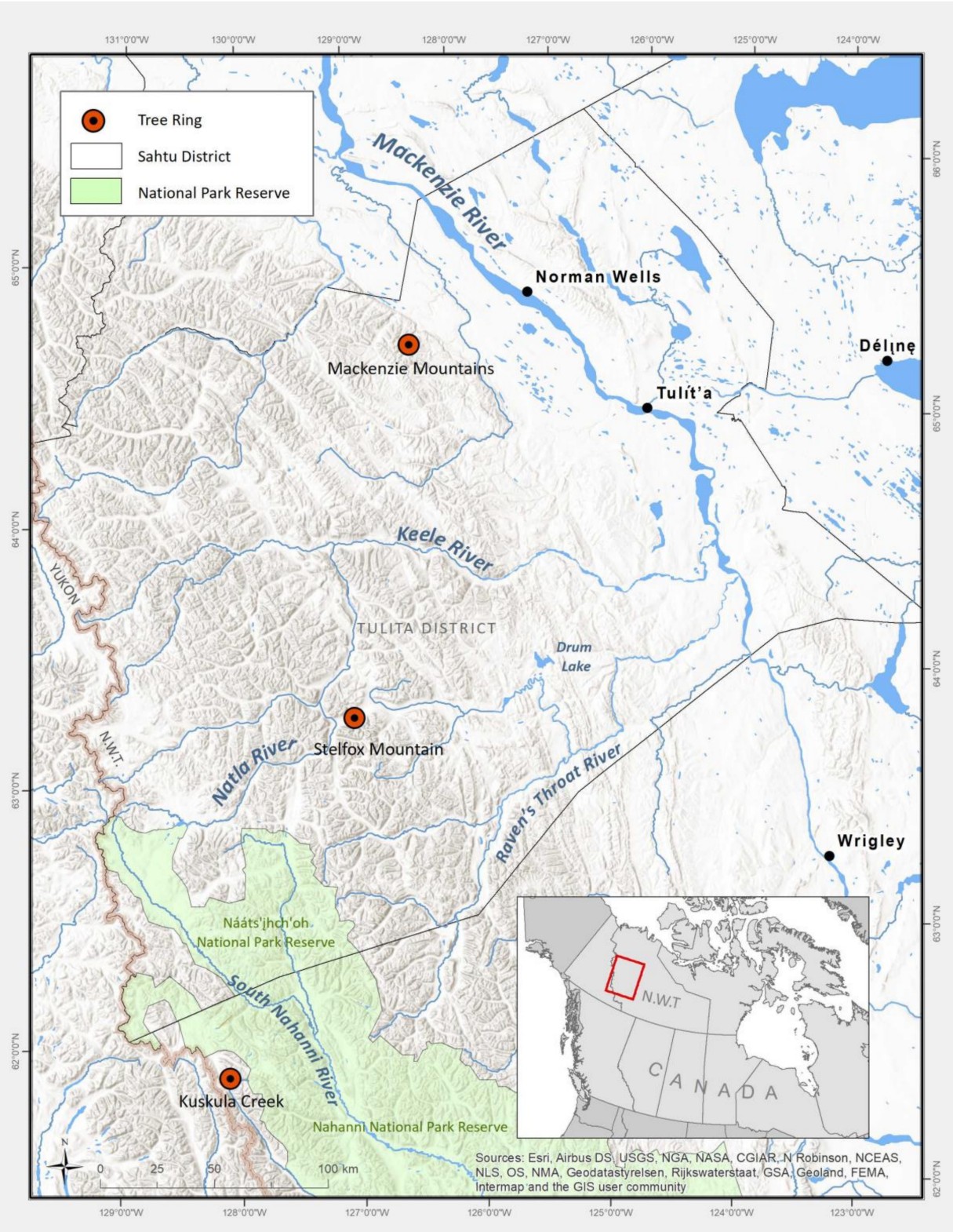

**Figure 4.** The location of the study site at the base of Stelfox Mountain in a remote section of the Mackenzie Mountains in the Northwest Territories. The closest towns are Norman Wells and Tulit'a, which are located on the Mackenzie River in NT. The two closest tree-ring chronologies to the study site are the "Mackenzie Mountain" chronology found north of the study site [19] and the "Kuskula Creek" chronology found to the south [20].

### 2.3. Standing Dead Wood

Twenty-one sets of cores from standing-dead trees (snags) were also collected to assist in potentially extending the 'living' chronology further back in time. In harsh mountain environments, standing snags can remain stuck in time with little to no degradation to the wood for centuries (e.g., [24]). If this is the case at the study site, then standing snags can be crossdated into the existing living chronology and extend the "anchored in time" master chronology back further. An extended chronology would greatly assist in bridging the presumed time gap between the living tree dated master chronology and the potentially much older archaeological wood from the fence system. Snags were sampled in much the same way as the living trees. In the 2016 field season, 21 samples were collected from the study site, and an additional 15 standing-dead trees were obtained in 2017.

### 2.4. Archaeological Samples

Archaeological wood samples were obtained from throughout the fence system (Figure 2). A walk-through survey of each of the three fences was conducted, and solid portions of fence line tree boles were identified as potential sampling locations. Duct tape was wrapped around the boles at approximately 1.3 m from the root collar (similar to the sampling height of the live samples). Then, an approximate 12 cm section was cut out of the logs with a chainsaw and then wrapped in plastic for transportation back to the lab for processing. In total, 89 samples were collected across the three fences. The individual cross-sections from the 89 samples each provided a "floating chronology", one whose rings are not yet assigned to any known calendar years. Once an archaeological sample's temporally floating measurements was placed in time through comparison with the living/snag master chronology, the sequence was added to the master chronology, adding sample depth to the master, as well as pushing the calendar-dated master further back in time.

### 2.5. Laboratory Preparation

Initial preparation differs between the cores and cross-sections. Increment cores from the living and standing dead trees were glued into slotted mounting boards, and labels from the field were transferred onto the boards. Once dry, the samples were stored and awaited sanding.

Stem sections were cut to a width of approximately 5 cm using a bandsaw. Widths varied when the wood's physical structure made it impractical, or unsafe, to cut at the preferred thickness. Basic steps were taken to ensure that each cross-section maintained its structural integrity, which for most solid pieces meant simply wrapping them several times with duct tape. In cases where material loss and wood damage were evident, but not severe, wood glue was added in stages over several days to solidify the sample and to minimize deformation or degradation during further processing. In a few instances, standard processes did not provide the necessary degree of structural integrity required for further processing the wood. In these cases, steps were developed and applied experimentally with low-expansion foam, prior to cutting, to help stabilize the archeological samples. In the end, only two of the 89 samples were deemed unsuitable for processing because of the poor physical condition of the wood.

Both cores and cross-sections were sanded using an up to ten-step process. The environment at Stelfox Mountain is such that trees tend to grow radially very little each year. This results in very fine rings with boundaries that are difficult to determine without a highly polished surface. This usually provides sufficient visibility for rings to be seen and accurate measurements to be taken. In this study, additional sanding was required for both the cores and the cross-sections. This was performed using grits from 600–1200, which was followed by mechanical buffing to achieve a fine polish.

A Velmex™ stage system was used in conjunction with a 60X microscope to measure ring widths to 0.001 mm, which were recorded using J2X software [25]. Two records were generated for each live tree/snag, an "A" and "B" sequence representing the two cores

taken from each tree. The archaeological cross-sections were also marked with an A and B path distinguishing those radii with the greatest number of rings present on each disc. The recorded living, snag, and archaeological ring-width sequences were cross-dated using both visual and computer-assisted methods to determine the end dates of each sample.

*2.6. Building a Master Chronology*

The signal homogeneity of the living- and snag-tree measurements were checked using the computer program COFECHA [26]. Potential problems reported by COFECHA were addressed on a case-by-case basis. The problematic series were either remeasured or visually crossdated to determine the source of the problem. First, the master chronology represented from only the living trees was constructed; then, once all problems were accounted for, the snag samples were added one at a time. Snags were initially placed as was suggested by the program COFECHA, and then, they were visually checked for a robust pattern match to the living chronology. Fifty-year segments were used for all matching, and therefore, inter-correlation values of the chronology needed to exceed the critical correlation level of 0.3281 in order to exceed the 99% confidence interval [18].

The two paths of ring measurements recorded from every archaeological sample were first crossdated with each other prior to crossdating with the master chronology. The goal of this step was to eliminate the uncertainty resulting from two paths of potentially unequal ages exhibiting a mismatched patten relative to each other. Agreement between the two series from one sample would result in a more robust statistical result when crossdated against the master chronology. Once those individuals with the most recent end dates were found and added into the master chronology, progressively older series of archaeological samples could be accurately crossdated and added into the overall master chronology.

Therefore, the master chronology, which began with the living trees, was extended further back in time as snags and archaeological wood were added. As the archaeological content of the master chronology increases, so does the sample depth of the dated master. The final master chronology is consequently comprised of living, standing dead, and archaeological ring-width information longer than is possible to produce using living trees alone. In all cases, the detrending and standardization of all samples and all chronologies used a standard 32-year spline [27] in program ARSTAN [28].

### 3. Results

The final Stelfox Mountain master chronology covers the calendar years from 972 to 2016 C.E. (Figure 4). COFECHA reported a series intercorrelation for the overall master chronology of 0.562, which was well above the critical correlation target of 0.3281 (Table 1). This demonstrates that although individual tree responses to environmental conditions varied widely, the community at large shares strong tendencies throughout time.

**Table 1.** Final master chronology statistics from the study. * Note the series intercorrelation is based on overlapping 50-year segments.

| Sample Type | Number of Trees in the Chronology | Chronology Length (Years) | Mean Series Intercorrelation * | Mean Sensitivity | Mean 1st Order Auto-Correlation |
|---|---|---|---|---|---|
| Chronology | 165 | 1045 | 0.562 | 0.240 | 0.746 |
| Live | 74 | 377 | - | - | - |
| Snags | 29 | 863 | - | - | - |
| Archaeological | 62 | 904 | - | - | - |

The living tree sequences ranged in ages from 49 to 376 years old, with 74 living trees from zones around the fences contained in the final master chronology (Table 1). Trees sampled to the north of the fences proved to be oldest, while those from the plateau proved to be much younger (Figure 2). Twenty-nine individual snags were included in the final chronology (Table 1). Three had end dates in the 1900s, while the oldest was crossdated into

the master chronology between 1149 and 1450 C.E. (Figure 5). The snags' contribution to the final sample depth of the site's master chronology was especially evident between 1750 and 1850 C.E. During this period, the number of living tree samples in the chronology steadily decreased (Figure 5). By 1750 C.E., living trees only contribute 10 series to the sample depth of the master chronology (Figure 5). The inclusion of snags in the site chronology served to almost double the combined contribution of archaeological and living series over this specific period (Figure 5). This became especially important as problems in crossdating were evident in this period, and many of the key narrow pointer years came from this interval (Table 2).

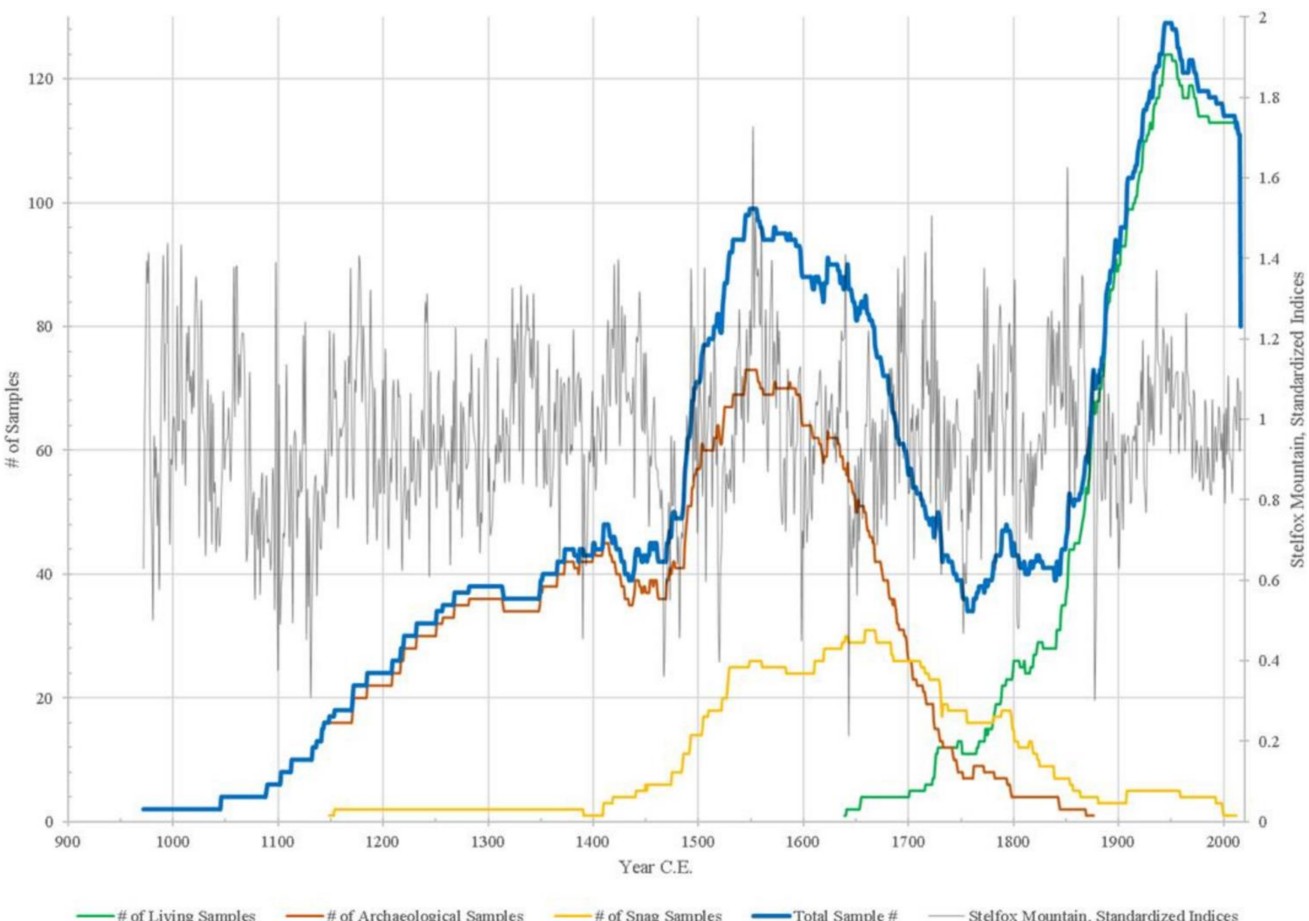

**Figure 5.** The master chronology (black) and sample depth of the overall chronology (blue), as well as the three components of the chronology. The live cores (green), the dead standing snags (yellow), and archaeological samples (red).

　　At its completion, 62 of the viable 87 archaeological cross-sections were successfully placed within the site's master chronology (Table 1; Figure 6). The average number of measured rings per cross-section is 214. Two distinct periods contain most of the archaeological end dates: 1420–1480 and 1580–1750 C.E. (Figure 6). Only two archaeological end dates fall in the 19th century calendar years (1876, main fence, 16LD059 and 1843, north fence, 16LD010). The earliest archaeological sample added into the master chronology had an end date of 1314 C.E (south fence, 16LD020); however, 19 samples could not be confidently added. An additional six samples could not be crossdated, as they were either much older than the master chronology or shared no robust pattern-matching sequences with the master.

**Table 2.** Crossdating marker years that note years of statistically narrow radial growth. These marker rings were used to help establish the overall pattern of growth within the master chronology between the overlapping live, snag, and archaeological samples.

| Stelfox Chronology Marker Years | | | | | |
|---|---|---|---|---|---|
| Year | Narrow Ring | Frost Ring | Year | Narrow Ring | Frost Ring |
| 1185 | X | X | 1721 | | X |
| 1205 | X | X | 1752 | X | |
| 1244 | | X | 1769 | X | |
| 1300 | | X | 1773 | X | |
| 1320 | X | | 1783 | X | |
| 1467 | X | | 1799 | | X |
| 1473 | X | X | 1803 | X | |
| 1504 | | X | 1804 | X | |
| 1507 | X | | 1805 | X | |
| 1519 | X | | 1817 | X | |
| 1520 | X | | 1819 | X | |
| 1550 | X | | 1850 | X | |
| 1598 | X | | 1877 | X | |
| 1623 | | X | 1898 | | X |
| 1648 | | X | 1904 | X | |
| 1710 | X | | 1973 | X | |

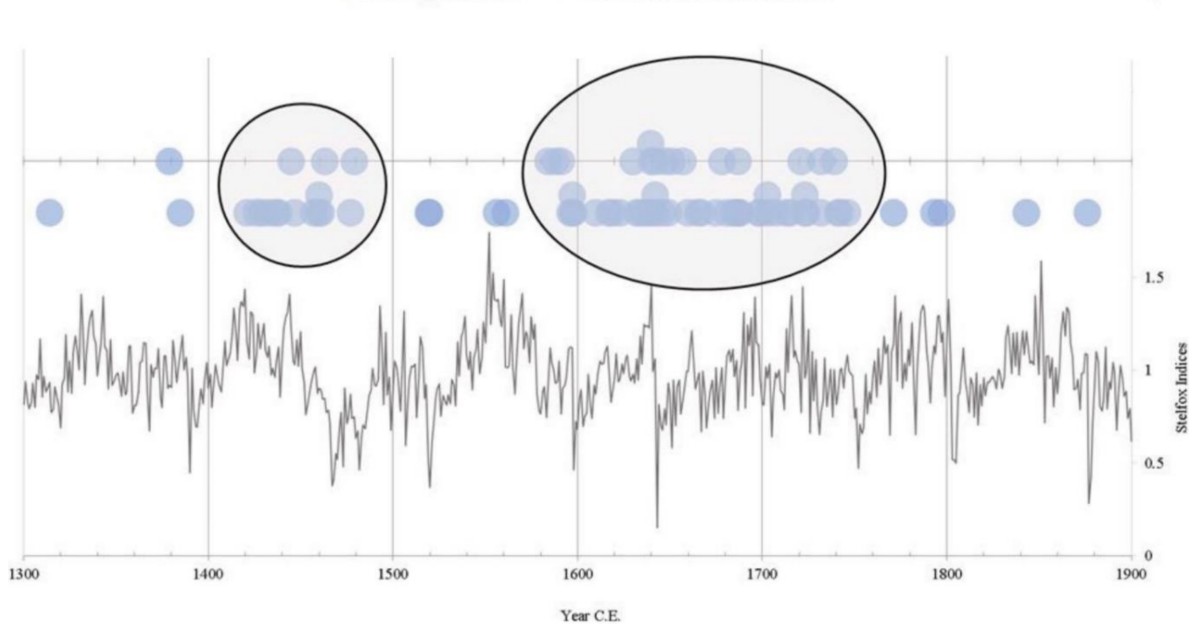

**Figure 6.** The time periods of the end dates for the 81 archaeological samples crossdated in this study are represented by blue dots. The end dates are spread over a period from 1314 until 1876 C.E., but most of the samples cluster into two time periods, 1421–1477 and 1597–1746 C.E.

The end dates of the 19 samples determined by crossdating that only partially fit with the master chronology were not added to the overall master chronology. For example, some of these cross-sections had portions of their sequences in their earliest years that did not crossdate well, which was often because of localized rot, missing rings, reaction wood, low-sample depth to match against, or a combination of one or more of these reasons. Since the entire measurement sequence could not be matched with a confidence above the 99% level, the data did not make it into the final master chronology (Figure 5). The data provided by their end dates were still very valuable though, and so they were still included in the final analysis of the spatial and temporal date clusters attributed to the individual

fences, if at least three overlapping 50-year segments could be crossdated against the master chronology (Figure 6).

## 4. Discussion

In the early stages of the chronology's development, several time periods provided significant obstacles for crossdating. The period between 1799 and 1805 C.E. was particularly problematic because of a probable extreme frost event (1799), which was evident in almost every ring sequence, followed by a period of extremely limited growth from 1803 to 1805 C.E. The growth between 1803 and 1805 C.E. was so stunted that most trees exhibited at least one locally absent ring between their two cores; in many cases, two years were missing, and in one instance, all three years were missing. These factors threatened to prevent extension of the chronology back into the 18th century. Repeated graphing and statistical analysis initially provided no conclusive indication of the actual sequence in this period.

High-resolution scanning was employed to attempt various methods of visual crossdating across this period. Cores with uninterrupted sequences and matching marker rings on either side of the early-1800s sections were required to bridge and eventually understand what happened in the affected years. Snag samples proved to be especially helpful in this process. Eventually, several cores were found to be free of a break in their sequences and shared the marker rings 1769, 1773, 1783, and then 1817, 1819, 1850, 1877, and 1904, on either side of the 1799 ring disturbance pattern (Table 2). Crossdating these specific marker rings allowed the site chronology to extend back into the 18th century and ultimately opened the entire chronology to wider development.

Since the chronology is from a remote location, a method of comparing the Stelfox chronology against other comparable chronologies was sought to verify the chronology's accuracy. The International Tree-Ring Data Bank (ITRDB) provided the two closest studies from the periphery of the Mackenzie Mountains. The "Mackenzie Mountain" chronology from ≈175 km north of Stelfox Mountain was made of white spruce (Jacoby et al., 2005), while the "Kuskula Creek" chronology was taken from ≈175 km south of Stelfox Mountain and was made from black spruce (*Picea mariana*) (Sauchyn 2008) (Figure 4). The raw data from the two sites were standardized and plotted against the Stelfox chronology using the same methods in this study (Figure 7) to determine if a general pattern of growth was shared between the three chronologies. The comparative data indicate a similar trend over most of the course of their 300-year overlap (all three chronologies) and a further 200-year overlap between the two white spruce chronologies until the "Mackenzie Mountain" chronology ends in the early 1500s.

It was expected that differences in individual years and potentially magnitude would be present between the three chronologies, but that their shared general pattern would be consistent. The only apparent difference was between 1820 and 1870 C.E., where the magnitude of the Stelfox chronology is higher than the other two sites. There is a noticeable decrease in the sample sizes of living trees at all three sites during the 1650–1750 C.E. period, indicating a shared set of difficult growing conditions throughout the mountain range. It is before this decrease in the sample size of the living trees that the archaeological wood sample size from the fences substantially increases. It may be worth exploring why this gap between the decrease in living and increase in archaeological wood exists as well as exploring Shúhtagot'ine oral history and the archaeological record for any other cultural corollaries during this period of potential climatic adversity. A significant period of poor growing conditions exists ≈1700 C.E., and therefore, the overall ecological growing conditions and a widespread death of trees may be reflected by the drop in sample size.

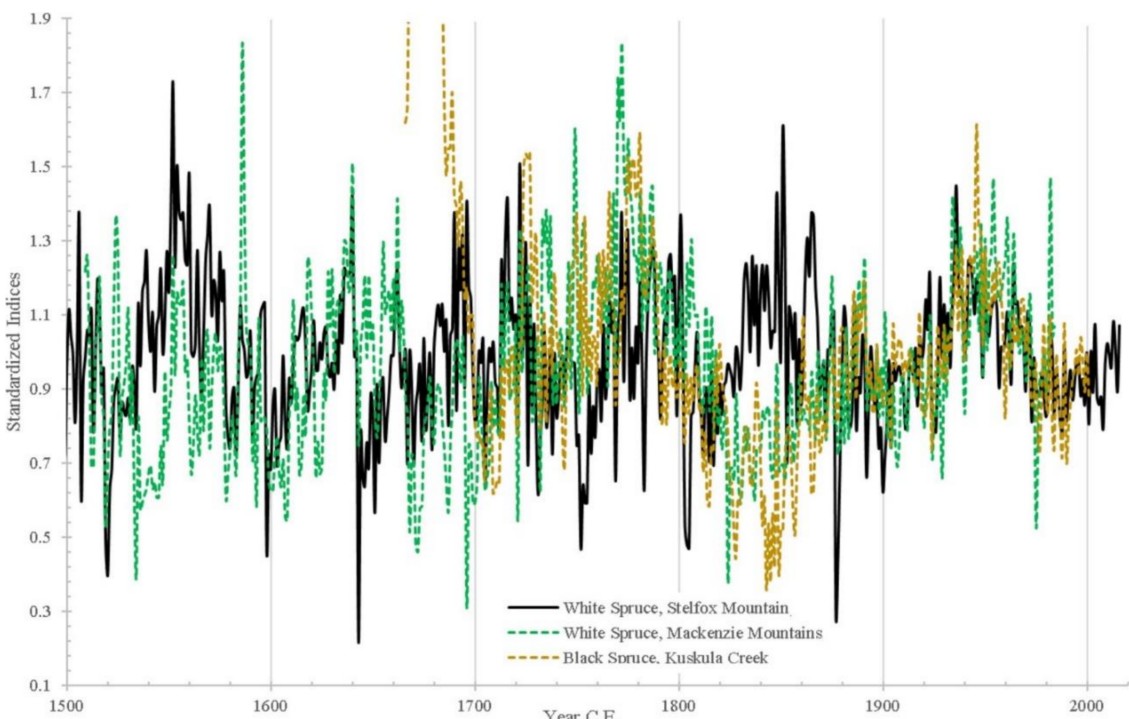

**Figure 7.** A comparison of the Stelfox chronology (black) and the two other closest chronologies available through the International Tree Ring Data Bank over their common interval. The three chronologies compare well both visually and statistically except for a short period in the early to mid-1800s.

With an apparent agreement between the three sequences, it appears that methods employed to circumvent problematic years in this study were successful, and that the Stelfox chronology will be an accurate and suitable chronology that can be used for dating archaeological fences and other suitable materials from a wide geographical region. With an agreement in ring patterns over a circular spatial range of 175 km, and with a 1000+ year chronology to use, many archaeological samples from a wide range of features should be able to be dated using the chronology developed in this study.

The successful crossdating of 81 pieces of archaeological wood reveals date clusters around 1420–1480 and 1580–1750 C.E. and indicates that trees killed as late as 1876 were added to the main fence structure. These data provide a general timeframe for the fence site and may indicate discrete periods of potential site construction and use, and/or that the people who built the fences made extensive use of standing deadwood and deadfall through time. This may be particularly important when interpreting the information in Figure 6. The crossdated snags indicate that standing snags can survive unaltered on the landscape for up to 500 years. The extended master chronology will facilitate the crossdating of additional fence samples to help refine the history of the fence structures.

## 5. Conclusions

This investigation reveals that Stelfox Mountain has been home to a well-populated coniferous forest for over a millennium. As a result of the extremely slow processes of decay in this harsh environment, living trees, snags, and cross-sections taken from the archaeological remains of the Moose Horn Caribou Fence were used to build an over one-thousand-year crossdated tree-ring chronology (972 to 2016 C.E.). The chronology is the first of its kind for dendroarchaeology studies in the middle of the Mackenzie Mountain region. It is the first chronology developed using dendroarchaeological methods on such a large-scale feature in the subarctic.

The threat of fire, or other events caused by regional warming, are a recognized danger to the archaeological resources of this site and others in the region. If these events come

to pass, it will be a significant cultural and scientific loss. Thus, even though the current project resulted in some minor harm to some of the wood in the fences, the chronology itself also has been identified as having the potential to combine with Shúhtagot'ine Traditional Knowledge and archaeological data to expand the understanding of the site and its cultural and environmental context. Bringing these empirical and cultural aspects together, to the benefit of each, presents a unique opportunity to explore past land-use change within this dynamic environment. The master chronology developed in this study may now also potentially allow the dating of other important cultural artifacts important to the Shúhtagot'ine within their traditional area of the central Mackenzie Mountains.

**Author Contributions:** Conceptualization, G.B., G.M., T.A. and C.L.; methodology, G.B. and C.L.; formal analysis, G.B. and C.L.; investigation, G.B., G.M., L.A., T.A. and C.L.; resources, L.A.; writing—original draft preparation, G.B.; writing—review and editing, G.S., G.M., L.A., T.A. and C.L.; visualization, G.B. and C.L.; supervision, G.S., C.L. and L.A.; project administration, G.M. and L.A.; funding acquisition, G.B., G.M. and T.A. All authors have read and agreed to the published version of the manuscript.

**Funding:** This research was funded by The Prince of Wales Northern Heritage Centre [2016-001], and student scholarship grants to G.B. by ERSI Canada GIS, Garfield Weston Award for Northern Research, and the Northern Scientific Training Program.

**Institutional Review Board Statement:** Not applicable.

**Conflicts of Interest:** The authors declare no conflict of interest.

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
