# Peer review of "Creating a Millennial-Long Chronology in Northern Canada: Dendroarchaeological Dating of the Moose Horn Pass Caribou Fence (KjRx-1), Mackenzie Mountains, NT"

_forests, doi:10.3390/f13020221_

Round 1
Reviewer 1 Report
Very good work in my opinion !
A sketch of the schematic structure of such wooden fence, ti be added to the work, should and would be highly desirable.
American wooden fences are quite dissimilar from European (and W EURASIAN) recent or subrecent ones. But "aboriginal " ones apparently dIsplay extremely intersting common structural traits, worthy of graphic documentation, for the benefit of ethnographuc research
Author Response
Reviewer #1 suggested we add a schematic of how the fence was built at our site. We have added this suggestion as a new Figure (Figure 3) at Line 75 of our revised document.

Reviewer 2 Report
Manuscript ID: forests-1558728
Type of manuscript: Article
Title: Creating a millennial-long chronology in northern Canada:
Dendroarchaeological dating of the Moose Horn Pass Caribou Fence (KjRx-1), Mackenzie Mountains, NT
Authors: Gary J.R. Beckhusen, Glenn Stuart, Leon Andrew, Glen MacKay, Thomas D. Andrews, Colin P. Laroque *
Submitted to section: Wood Science and Forest Products,
Title: Dendrochronology: An Interdisciplinary Approach to Assess Wooden Cultural Heritage Worldwide
General Comments: The authors describe a perfect application of dendrochronological techniques to reconstruct the construction history of a landscape significant archaeological feature. Speaking on to the application of tree-ring dating techniques I find no seriously objectionable representations or findings. What “correction/clarifications” I do find I
have embedded here at the bottom of this document and in the edited MS word version accompanying this review. Regarding the presentation I suggest the authorscommit the current ms to some more careful editing, be aware of tense and plurals. As for the thesis of this article I only have one serious concern and that pertains to the origin and composition of wood in the fence. I find it less likely to believe the original builders would have purposely felled trees for fence construction but more likely harvested nearby and local dead snags and fallen timbers, dragging them to
where needed. Which raises the question, what evidence is there that any of the trees used to construct any portion of these fence lines were felled/killed by people?? If such evidence exists it must be provided. Add to this the strong possibility that the last ring formed before a fence line tree died, may be missing, due to erosion and sapwood decay, puts wider constraints on the confidence interval of construction dates. These variables should be more clearly mentioned in the text. What is the rate of spruce decay in the arctic? I agree that is a question for another study, but nevertheless its existence should be recognized for we know all organic material decays (see: Matthiesen et al., 2014, Degradation of archaeological wood under freezing and tha wing conditions—effects of permafrost and climate change,
Archaeometry, Sellin A., (1994) Sapwood–heartwood proportion related to tree diameter, age, and growth rate in Picea abies CJFR)
Regarding the interpretation of the results and fence line functions I have two thoughts. First, if the author’s theory regarding these fence lines is to “corral” animals into a kill zone then the North Fence presents a bit of a conundrum. Is it possible the space created by the North Fence, relative to the Middle Fence, provides a means of dissecting the herd, possibly separating cows from bulls, young from old ? Just a thought.
Specific Comments:
Fig.2. This is a map therefore it must contain location coordinates in latitude and longitude, preferably in decimal degrees.
Line 157. In dendrochronology, this "pattern-matching" procedure is called crossdating.
Line 167. I think it would be helpful to point out much earlier in the text that the fence line material is largely composed of branches and whole tree strems that 1) have not been artificially modified in any way, ie, still retain their natural condition, barring erosion and decay and 2) are not all asuumed to have been puposely felled by humans to build the fences. Some in fact may have died much earlier and were either standing snags or laying about long before being incorporated in the fence.
The latter point is criticle for when it comes to interpreting the sequence of deathdate pulses/cluster in the final chronology.
Lines 199-201. Please don't write this for it is not true (see: Stoke and Smiley 1968, Krusic et al 1987) Generally speaking, there is no “general protocol/sequence of sanding papers for preparing samples. The goal of sanding is to produce a surface fine enough to see clearly every ring boundary, as defined by the last latewood cells in every ring. How, and with what numerical progression of papers thid is accomplished, is irrelevant. What is relevant is the procedure ends with a "highly polished" surface. For example, Jacoby et al 1985 -2005 produced all their high lat. boreal forests chronologies using a progressive regime of 240, 400, 600 and 800 grit
papers. I found this progression works well as a starting point for dealing with most conifer species around the world. It is fine to describe the steps you have taken in detail, but please be careful with extending what you have found to beyond your experience.
Line 216. Only Holmes 1983 is sufficient here. If an additional ref. is required, one demonstrating program usage, then cite; Grissino-Mayer, HD. · 2001 Evaluating Crossdating Accuracy A Manual and Tutorial for the Computer Program COFECHA.
Tree-Ring Research, 57, 205-221.
.
Line 244. So far we have not hear anything about the detrending/standardization method used to produce this chronology (Fritts 1976). Consequently, I can only assume it is the default 32yr spline (Cook and Peters 1981, Holms 1983, GrissinoMayer, HD, 2001). For dating purposes this flexible a detrending spline may be fine (hence its harcoded default value) but for the purpose of climate interpretation one could argue using this flexible a spline may remove valuable low-frquency information attributable to climate.
Table 1. Please add EPS and Subsample Signal Strength, %missing rings (if none ok), (see: Buras, A. (2017) A comment on the Expressed Population Signal.
Dendrochronologia 44:130-132., Cook, E. R. and Kairiukstis, L. A., editors
(1990) Methods of Dendrochronology: Applications in the Environmental Sciences. Springer). If your data are stored in decade format, the default output file format from program Measure J2X and input format for program COFECHA, then you can easily run them through program ARSTAN
(https://www.geog.cam.ac.uk/research/projects/dendrosoftware/) to compute these values. Be sure to change the first detrending from the default age-dependent spline to a fixed 32 year spine to mimic your COFECHA results.
Lines 298-301. This statement begs the question, if a sample could not be
confidently crossdated, how could that sample's "end-dates" be determined and therefore used in the cluster analysis?
Lines 318-319. I think of equal interest are the number of radial sequence that DID contain the 1802-1805 rings, which permitted you to correctly identify the later 18th century marker rings. Again, please include the number or % of missing rings in the master. Even a figure or table illustrating the the year and number of missing, frost, and locally absent rings would be valuable. This last suggestion may not be very pertinent to your archaeological effort, but certainly interesting to others working to
understand past local climate conditions.
Lines 350-353. Is it not plausable that the poor conditions during the late 18th century had an affect on reproduction/seedling survival and thus the seemingly much lower average age of all living trees?
Lines 355-356. So, are you going to make the crossdated dated used to build your master chronology publically available? Much like how you downloaded the Mackenzie and Kuskula data for your study would others be able to do the same with your data once this article is published? I would applaud you for doing so.
Lines 356-357. I wonder about this statement. One could support it by proving the results of a COFECHA run where each of the other two chronologies is tested against yours. Just like you used cofecha to verify the dating of your individual samples, you could do the same with these two chronologies. It may in fact turn out to be that some of the low-frequency descrepencies seen in fig. 6 are the result of different standardizations used to produce each chronology. If you have access to the raw data used to produce the Mackenzie and Kushkula chronologies you can use COFECHA or ARSTAN to produce a high-frequency/32 year spline chronology of each. In COFECHA, selecting Option 6 in the main program menu saves the "dating" chronology produced in 1 of 3 formats. If you save the chronogies produced in the default ITRDB format you can run them as “undated” measurements against your dating chronology to produce a suite of correlation stats.
Lines 365-366. YES. This is an important point to consder when interpreting the information in fig.5.
Please see the attachment.

Author Response
As indicated in the tracked changes of the resubmission, we added more of an explanation of the wooden materials at the site, and the axe cuts on some of the beams in lines 58-65. This eliminated the needs seen later in the discussion of the manuscript to address the use of axes at the site by Reviewer #2.
Reviewer #2 made a number of editorial suggestions with the wording throughout the manuscript in tracked changes. We accepted almost all of their comments. Specifically, we did not accept a comment that they made in the Table 2 caption (line 280), where the reviewer added the word “years”, and this pushed its use to three times in one sentence. We rejected this, and instead we reworded the sentence to reduce duplication in the use of the word.
Reviewer #2 added some reference suggestion throughout the manuscript, and we incorporated them, except we did not incorporate the Grissiono-Mayer reference on line 224. We also added one reference, Holmes et al. 1986 in lieu of the suggested addition of Holmes 1983. We feel that the Holmes et al. 1986 reference is actually better and is more widely used in the literature for the suggested purpose that Reviewer #2 was trying to make us make the change. We did like the suggested addition at this location though.
Reviewer #2 asked us to perform an EPS test on our data set to assist readers who are interested in dendroclimatology (line 260). We agree that this can be done easily, and that we are able to do it, but we do not wish to add the test to the manuscript. EPS is a signal strength test that is important when trying to understand the signal strength of hindcast relationships etc. when conducting dendroclimatological modelling. In regard to this paper though, we feel this will only detract from the archaeological special issue, as this test is never used in dendroarchaeological applications. We therefore did not include the suggestion in our resubmission.
As an aside, Reviewer #2 asked if we are going to share the data in the IRTDB, and so far we have not received permission from the Shúhtagoťine people to do so, as they are the owners of the data. We hope to. When this is accomplished, it will also allow each person to play with our data set as Reviewer #2 asks when they ask us to perform an analysis on how many samples did contain the full 1803-1805 ring sequence and how many had null values within this triplet of rings. This assessment would be interesting to the reviewer, but we think these types of tangents start to derail the point and direction of our manuscript. We therefore did not run this side analysis suggested by the reviewer on line 337.
Another aside that Reviewer #2 had, was they asked us to run a duplication test that they thought up in a comment on line 378. We did not do this as the other chronologies were not built by us, and procedures may or may not have been similar. We at least knew what we did to arrive at our chronologies, and so we left it as the individual tests that we performed and what was described in the manuscript.
